# Innovative Closed-Loop Recyclable Bio-Based Composites from Epoxidized Waste Flour and Recycled Carbon Fibers

**DOI:** 10.3390/polym14183878

**Published:** 2022-09-16

**Authors:** Francesca Ferrari, Gloria Anna Carallo, Antonio Greco

**Affiliations:** Department of Engineering for Innovation, University of Salento, Via Arnesano, 73100 Lecce, Italy

**Keywords:** bio-based composites, closed-loop recycling, epoxidation

## Abstract

Epoxy-based composites are designed for long-lasting applications, though their wide use is in contrast with their poor recyclability, which poses serious end-of-life issues. In order to reduce their environmental impact, precursors derived from fossil fuel based raw materials should be replaced with eco-friendly sources. This can be attained by using naturally derived epoxy matrices, or by finding a suitable solution for recycling at the end of life. In this paper, both strategies were analyzed, by replacing traditional monomers with epoxidized waste flour (EWF), an innovative bio-precursor derived from the organic waste stream, and a cleavable hardener, which allowed the recyclability of the matrix. The recyclable matrix was reinforced with recycled carbon fibers, derived from pyrolysis. DSC measurements were carried out in order to optimize the curing steps of the matrix, then flexural tests were performed in order to evaluate the mechanical response of the composite. A green recycling procedure was then investigated, which involved the use of non-toxic solvents and mild working conditions, and allowed recovery of the matrix while still preserving the properties of the carbon fibers. The components obtained after recycling were analyzed by FTIR analysis, which revealed the presence of the epoxy ring on the recycled waste flour. Hence, recycled waste flour was again used as a precursor and mixed with the cleavable hardener, thus, obtaining a closed-loop recycling.

## 1. Introduction

Composites obtained from epoxy matrices are light and long-lasting materials, planned for a wide timeframe (20–30 years), and are used in several industrial sectors often when high performances are required, e.g., aerospace or automotive. Currently, most of these products on reaching the end of their life will be disposed [1], as epoxy-based composites cannot be recycled by traditional methods, since the crosslinks of the thermoset matrix do not allow remelting and reshaping. Therefore, an appropriate system to separate the matrix from the fibers must be investigated.

The traditional options for discarding thermoset composites are incineration and landfill. To date, landfill has mostly been used since it is a relatively cheap disposal route [2], but it has low efficiency and economic rates. Additionally, landfill disposal of composite structures has a very high environmental cost since it significantly contributes to the increase of land and air pollution [3]. The other traditional route is incineration, where waste combustion occurs, producing a high amount of ash and harmful powders, along with chemical slugs.

Recycling should be the preferable option for dealing with composites at their end-of-life [4], although only a few recycling technologies can be applied to thermoset composites: mechanical, chemical and thermal recycling [5].

Mechanical recycling consists of shredding and grinding processes which reduce waste into recyclates; these products aim to return in the processing route as fillers or reinforcement. Unfortunately, grinding process can negatively affect the original properties of the fibers (mechanical strength and stiffness), and their interaction with virgin matrix could be difficult without proper pre-treatment. These considerations, in addition to important costs still associated with the processes, limit the efficiency of this method, which should be improved, for instance by increasing the materials rate (with a consequent reduction in energy demands).

Thermal recycling allows the separation of fibers from the matrix, often achieved by pyrolysis of the matrix. High temperatures are applied to the composites in order to extract and recover the fibers, while the matrix is usually disposed. The main disadvantages are, first of all, the complete loss of the matrix, and secondly, that the quality of recovered fibers is usually very poor, since a part of the char [2].which is formed due to high temperatures and oxidation is deposited on their surface; removing the char fraction from the fiber surface involves additional damage of fibers. Moreover, thermal energy consumption is very high, reducing the environmental efficiency of the process.

Finally, chemical recycling leads to the conversion of polymers to monomers/oligomers via chemical routes. A preceding grinding is also sometimes performed, in order to facilitate the chemical reactions on large structures. The main drawback associated with this method is the use of chemical agents and solvents which significantly increase the environmental impacts of the process [2]. Recent studies have focused on the introduction of water, alcohol or other solvents without environmental concerns; the solvolysis method has been proposed for epoxy-based CFRP using molten KOH [6], but is still at laboratory scale, with further improvements and refinements required.

Therefore, despite the proven reduced impacts of recycling versus other strategies for composite disposal, the recycling methods currently available have some significant drawbacks [7], such as high monetary and energy costs, a decrease in the performances of the recovered products, and the possible use of toxic solvents. Moreover, as discussed in [8], their technological readiness level (TRL) is relatively low, which is mainly due to the poor process yield [9].

This work focuses on alternative chemical routes for the recycling of carbon fiber reinforced composites, which allows the reprocessing of the epoxy thermoset resin and preservation of fiber properties. In particular, in this work, the composite was recycled by a recently derived method, which involved its de-polymerization by using non-toxic solvents (water and acetic acid) and a cleavable hardener developed by Connora Technologies [10].

An additional environmental limitation of currently used epoxy systems is the use of petroleum-based raw materials for their synthesis. In order to overcome the limitation of synthetic materials, different natural sources have been investigated, and subsequently used, as precursors for epoxy resins. For example, Omonov et al. [11] developed a new bio-derived thermoset using epoxidized canola oil (ECO) and phthalic anhydride (PA) as a curing agent; once the thermal properties of the material had been investigated, they proved to be suitable for composite applications. In 2017, Sudha et al. [12] produced different epoxidized castor oil/DGEBA blends at various wt %, however, using triethylenetetramine (TETA) as the hardener: the rheological tests on the produced resin resulted in a lower initial viscosity than the commercial benchmark. Another natural-based precursor object of investigation is cardanol, which was successfully employed by Atta et al. [13] in the production of both bio-epoxy and bio-hardener suitable for marine applications. Additionally, Darroman et al. [14] introduced cardanol and sorbitol for the formulation of bio-based epoxy blends which exhibit interesting properties for coating applications. Finally, vanillin has gained interest in recent years as a precursor of bio-based thermosetting resins preparation: Shibata et al. [15] developed a bio-based aromatic epoxy resin starting from a by-product of vanillin, then cured with phenolic compounds, while Nikafshar et al. [16] produced a bio-based renewable epoxy resin from vanillin with outstanding properties, which could compete with standard DGEBA. On the other hand, it must also be observed that most of the developed bio-epoxies have been produced from primary resources, which involve the subtraction of raw material which could potentially be used in other applications (for example, in the food chain).

Epoxidation of waste flour, already studied in previous works [17], uses an innovative technology with a combination of UV and ozone, with positive reflections from an environmental impact point of view [18]. Moreover, Cicala et al. [19] successfully added a bio-based cleavable hardener to natural-based epoxy monomers to obtain recyclable composites by a resin infusion process.

On this research line, Ferrari et al. [20] tested several mixtures containing waste epoxidized commercial flour natural-based epoxy and cleavable hardener in order to optimize the performances of the natural-based epoxy resin and its recyclability.

These latest experiences prove that the use of organic waste as a precursor to the production of thermosetting polymers, with recycling features, is a valuable alterative to standard processing routes and raw materials for composite production and recycling, in order to reduce environmental impacts [18].

In this paper, for the first time, the recycling route developed for the bio-based epoxy matrix was applied for the recycling of bio-composites obtained by the addition of recycled carbon fibers. The proposed approach allowed recycling of both the matrix and fibers, almost entirely preserving the mechanical and physical properties of the matrix and reinforcement.

## 2. Materials and Methods

Waste flour (WF) was obtained from the processing waste of pasta factories. Epoxidized waste flour (EWF) was obtained by contemporary exposure to UV radiations and ozone for 5 h, following the method reported previously in [17] and in a patent [21]. A medium pressure Hg UV lamp (UV HG 200 ULTRA), with a radiation intensity of 9.60 W/mm^2^ on the surface of the samples was used for the waste flour treatment.


**Resin samples with epoxidized waste flour (EWF)**


The composition of all the produced samples is reported in Table 1. The first sample, EWFm, was obtained by mechanical mixing of epoxidized waste flour (EWF) with 16.7% by weight of Recyclamine^®^ R101, a natural-based cleavable curing agent by Connora Technologies.

The amount of amine was chosen based on the results reported in a previous work, where it was shown that the amount of amine of 16.7% allowed obtainment of the best mechanical and thermal properties [20].

The second sample, CFcomposite, was produced by adding 7.69% by weight of recycled carbon fibers (obtained by pyrolysis according to [22]) to the epoxy system, obtained by mixing EWF and the Recyclamine^®^ R101 curing agent. However, in this case, the high viscosity of the matrix produced with EWF alone did not allow for an efficient impregnation of the fibers, and consequently, very poor properties were obtained. Therefore, in order to improve the impregnation of the fibers, the EWF was mixed with a commercial bio-based epoxy. Polar Bear^®^ from R*Concept is a partially bio-based commercial epoxy system (>19% of bio-content) specifically tailored for composite processing.The composition of the CFcomposite sample is reported in Table 1.

After recycling, the recovered matrix was again mixed with the same amount of amine, for the production of the recycled epoxy sample, rEWFm.

In addition to the samples reported in Table 1, the uncured EWF and recycled EWF are labeled as EWF and rEWF, respectively.

All samples reported in Table 1 were obtained by mixing the proper amount of the components in a HAAKE Rheomix (T = 20 °C; speed rotation = 60 rpm; t = 20 min), while the curing process was performed using a compression molding machine P7/91 by Campana s.r.l. (t = 2 h at T = 120 °C; closure pressure = 50 bar).

### 2.1. Recycling Procedure

The recycling procedure, as optimized in a previous work for the matrix system (whose composition is reported in Table 1, with the sample name EWFm) [20], was applied for recycling of the composite and involved two steps: (a) the composite sample solubilization in a solution composed of 50% glacial acetic acid and 50% distilled water, and (b) the precipitation after the addition of a basic coagulant, e.g., sodium hydroxide. Figure 1 shows a block-flow diagram describing the recycling process.

When recycling the composite, immersion in the acetic acid solution involved cleavage of the epoxy bonds, and precipitation of both EWF and carbon fibers: Figure 2a shows the precipitation of EWF on solvolysis, while Figure 2b shows the recovered carbon fibers, after drying. Therefore, the solution, which still contained dissolved commercial epoxy, was filtered and then dried at 40 °C until weight stabilization, for recovery of EWF and carbon fibers (CF). The yield of EWF recovery was 100%.

Afterwards, in order to also recycle the commercial bio-epoxy, sodium hydroxide was added dropwise to the acetic acid solution by following the procedure reported in [20]. In Figure 2c, a picture of the Polar Bear^®^ resin obtained upon addition of the basic coagulant is shown; this procedure allowed for the recovery of the commercial bio-epoxy, with a recycling yield of 96%.

Hence, the overall recycling effectiveness of the composite samples, by taking into account both EWF and Polar Bear^®^ epoxy resin, was 98%.

### 2.2. Methods

Differential scanning calorimetry (DSC) analysis was performed on a Mettler Toledo 622 (Mettler Toledo, Greifensee, Switzerland). Two scans were performed: a first heating from 25 °C to 130 °C (at 10 °C/min) followed by cooling at 10 °C/min back to room temperature. The second heating scan, again performed at 10 °C/min, was used to measure the relevant transition of the epoxy samples.

Fourier transform infrared spectroscopy (FTIR) analysis, performed with a FT-IR Jasco 6300 spectrometer(JASCO Corporation, Tokyo, Japan), was used to assess the presence of epoxy groups after UV/ozone exposure. Infrared spectra were recorded in the wavelength range between 400 and 4000 cm^−1^, 128 scans, and 4 cm^−1^ of resolution, by using a germanium round crystal window.

Dynamic mechanical analysis (DMA) was performed on samples before and after recycling by using a strain-controlled Rheometrics ARES rheometer (Rheometric Scientific, Piscataway, NJ, USA), with torsion geometry, increasing the temperature from −25 to 150 °C at 2 °C min^−1^.

The flexural properties of each cured sample were measured using a dynamometer, Lloyd LR5K, according to ASTM D790 (ASTM D790-17, 2017) (three points bending with specimen dimension: 80 mm × 10 mm × 4 mm). Five replicates were performed on each sample.

Scanning electron microscope (SEM) analysis (ZEISS EVO LS10, Germany) was performed on once-recycled carbon fibers, *1R* (obtained by pyrolysis and used in composite production), and twice-recycled carbon fibers, *2R* (recovered after composite solvolysis). Parameters used were: EHT: 20 kV; WD: 11.5 mm; magnitude: 10,000× (fibers diameter and surface structure).

Single fiber tensile tests were performed on an ARES Rheometer (Rheometric Scientific, Piscataway, NJ, USA.) on *1R* and *2R* carbon fibers, according to ASTM D3379-75 (gage length: 40 mm; extension rate: 0.005 mm/s). Ten replicates were performed on each sample.

## 3. Results and Discussion

### 3.1. DSC

Figure 3a shows the DSC curve of the EWFm sample during the second heating scan. The Tg value detected for the analyzed sample, 94 °C, was roughly the same as observed in a previous work [20]. In [20], different amounts of bio-amine were tested, proving that 16.7% was the ratio which provided the higher Tg value, almost equal to Polar Bear^TM^-Recyclamine R101^®^ systems. On the other hand, it was also shown that a further increase in the amount of amine could result in a decrease in glass transition, due to the plasticizing effect of the amine in the epoxy network.

Figure 3b shows the DSC thermogram for the rEWFm sample, which was characterized by a Tg of 56 °C. After the recycling by solvolysis, the EWF retained a certain degree of epoxidation and, even if the Tg of the material was decreased compared with the pristine EWFm system, the recycled EWF still proved to be suitable for use as a bio-epoxy matrix.

The DSC curve of the CF composite is reported in Figure 3c: two different Tg were identified (i.e., 69.9 °C and 133.6 °C). According to the results obtained in a previous work [20], the two different Tg values could be attributed to the partial miscibility between the EWF and the commercial bio-epoxy, which involved the formation of an interpenetrating polymer network, composed of two distinct phases, each with its own Tg, which was expected since there are both Polar Bear Epoxy^®^ and EWF epoxy systems.

### 3.2. FTIR

FTIR analysis on the EWF and the commercial epoxy was performed to assess the presence of epoxy bonds in the different steps of curing/recycling.

Figure 4 shows the FTIR spectra for the EWF before curing, highlighting the typical signals of the epoxy ring at 1260, 890 and 827 cm^−1^. After curing (EWFm sample), the peaks due to the epoxy rings disappeared, confirming epoxy curing reaction. Furthermore, waste flour recovered after immersion in the acetic acid (rEWF) solution, again showed the presence of the typical peaks of epoxy rings.

Due to the presence of epoxy rings after the recovery, it was considered that rEWF could potentially be reused as a precursor for the production of bio-composite; therefore, the recovered flour was again mixed with the same amount of cleavable amine and the cure process was carried out at 150 °C for 2 h, for the production of rEWFm.

Figure 5 shows the FTIR spectra of the Polar Bear^TM^-based systems. In contrast with what was observed for EWF, the commercial bio-epoxy did not show any signal related to the presence of an epoxy ring both after curing and after recycling; this indicated that, before recycling, the correct stoichiometric ratio had been used to mix the epoxy precursor and the hardener, so a complete cure without residual epoxy rings was reached. Additionally, the absence of signals relating to epoxy rings on the precipitate obtained from the recycling process indicated that the system could not be used again as an epoxy precursor [4].

### 3.3. DMA

The production of an epoxy thermoset starting from rEWF, for EWFm and rEWFm samples is described below. For both materials, there were three regions: a glassy region characterized by very high storage modulus >1 GPa; a glass transition region, where the storage modulus could decrease by a factor of 10–100; and a rubbery plateau region with a stable storage modulus (G′), proportional to the cross-link density.

The Tg value of the systems was calculated from G′, by considering its inflection point. In Figure 6, Tg values of 66 °C and 42 °C were calculated for EWFm and rEWFm, respectively. Both values were lower than those identified by DSC analysis but confirmed a reduction in the Tg of EWF-based systems after the recycling process: this evidence was related to both a reduction in reactivity after solvolysis, and to a variation of the epoxy group contents. This latter aspect could be the starting point of further investigations, oriented to a better tailoring of the amine content, based on the effectively quantified residual epoxy groups post-recycling.

Figure 7 reports the G′ curves of the Polar Bear^TM^-based system. The graph reveals that a thermoplastic polymer was obtained after recycling. In particular, the black curve in the graph confirms that a decrease in G′ due to the glass transition was only detected for the thermoset Polar Bear after curing. The glass transition temperature, calculated by using the inflection point method, was about 75 °C.

The red curve in the graph, which refers to the recycled Polar Bear^TM^, describes different regions: first, the glassy region with a lower storage modulus than the thermoset Polar; secondly, the glass transition, with a decrease in the storage modulus until the rubbery plateau; and finally, a further decrease in G′ at 125 °C, due to the melting of the polymer. In contrast to the curve of the cured thermoset, where nothing happened after the Tg until the sample began to degrade, the presence of melting after the rubbery plateau indicated that the material was a thermoplastic.

### 3.4. Flexural Test

Table 2 reports the flexural properties of the tested samples. The results, reported in Figure 8, show that the mechanical properties of the rEWFm sample were comparable to those in [20]. In a recent work [23], a comprehensive comparison of different commercial bio-based epoxies showed a modulus in the range of 2.2–3.4 GPa and a strength in the range of 48–75 MPa, which are values much higher than those found for the EWFm system. However, in comparing EWFm and other commercial systems, the fact must be noted that all the commercial systems were actually composed of a blend with oil-based epoxies. The addition of bio-based epoxies to oil-derived epoxies involved a significant decrease in the mechanical properties; for example, in [12] it was shown that the addition of 50% epoxidized castor oil (ECO) to DGEBA involved a decrease in the tensile strength from 70 to 18 MPa, which was a value very close to that found for the EWFm system, which is, however, fully bio-based.

The addition of carbon fibers involved an increase in flexural strength and flexural modulus of about 2.5 times, due to the reinforcing effect of the recycled carbon fibers.

Flexural tests performed on rEWFm showed lower stiffness and strength compared with EWFm, which confirmed the results from DSC (i.e., reduction in the glass transition) and DMA (i.e., lower G′).

Globally, the overall results proved the retained epoxidation after recycling and a good re-usability of the material, but a proper calibration of amine content is expected in order to improve mechanical performances.

### 3.5. SEM

Scanning electron microscope (SEM) analysis allowed measurement of the diameters of both *1R* and *2R* carbon fibers, needed for mechanical characterization, as well as observation of the fiber surface after the recycling process. In Figure 9, typical SEM images for *1R* and *2R* fibers are reported. As can be observed, the surface of *1R* fibers appeared more uniform with a very smooth surface, whereas some roughness was observed for *2R* fibers. In Table 3, the fiber diameters measured by SEM for *1R* and *2R* carbon fibers at 1000×, are reported. As can be observed, *2R* fibers were characterized by an average diameter which was about 10% higher than *1R* fibers. The diameter increase, and the higher surface roughness for *2R* fibers can be explained by assuming that the composite recycling process left some residual matrix on the fiber surface.

### 3.6. Mechanical Tests on Recycled Carbon Fibers

Tensile tests on a single filament of *1R* and *2R* recycled carbon fibers, are shown in Figure 10. Additionally, mechanical properties obtained from the single fiber tensile test on *1R* and *2R* recycled carbon fibers, are reported in Table 4. Despite the recycling process, *2R* carbon fibers retained their integrity and structure (as observed by SEM analysis) and most of their mechanical performances, which resulted as slightly lower than those of *1R* recycled carbon fibers.

Initially, the mechanical properties of *2R* fibers were obtained by considering the fiber diameter value (6.908 μm) as measured by SEM. However, during tensile testing, fibers and matrix left on their surface were in iso-strain conditions. Therefore, due to the much lower stiffness of the matrix compared with carbon fibers, it was possible to exclude the stress that was acting on the matrix. With this assumption, the diameter of the fibers measured by SEM analysis on *1R* fibers should also be used for the analysis of the tensile data on *2R* fibers. The data reported in the last row of Table 4 clearly showed that neglecting the presence of the matrix on the surface of the fibers led to much higher estimated properties for *2R* fibers, which had substantially the same properties as *1R* recycled fibers; this indicated that the recycling process did not alter the mechanical performances of the fibers and allowed their re-use in the processing route, with positive reflections on environmental impact.

## 4. Conclusions

In this work, the recyclability of bio-based composites was examined. To this purpose, a bio-based epoxy matrix composed of a mixture of EWF and commercial system was mixed with a proper amount of cleavable amine (Recyclamine^®^ R-101) and recycled carbon fibers.

The use of a cleavable amine allowed for a complete recovery of the bio-based matrix by immersion in an aqueous solution of glacial acetic acid.

FTIR analyses showed that, after recycling, rEWF was characterized by the presence of epoxy groups, whereas the commercial system did not show any relevant peak related to epoxies. Therefore, rEWF was mixed again with a proper amount of cleavable amine and cured under the same conditions as EWF.

DMA analysis on rEWFm samples confirmed the production of a thermoset material, although a reduction in both glass transition temperature and mechanical properties were observed, compared with the pristine EWF sample. However, the recycled epoxy rEWFm still showed relevant mechanical properties.

The proposed recycling process also allowed recovery of the carbon fibers with irrelevant property degradation. SEM analysis showed that the carbon fibers obtained after the recycling of the composite were characterized by a diameter increase and a higher roughness, compared with carbon fibers used for the composite production. This suggested that some matrix was still present on the surface of the fibers, which, however, did not involve any loss of mechanical properties.

Future developments will involve the curing optimization of the rEWF matrix, by a precise quantification of residual epoxy groups after EWF recycling and a better tailoring of the bio-amine content, in order to improve the mechanical performances of the material for composite applications. Finally, a life cycle assessment (LCA) on developed materials and methods will be carried out, in order to enhance the environmental savings of the recycling process and provide significant hotspots for further improvements in the production route.

## Figures and Tables

**Figure 1 polymers-14-03878-f001:**
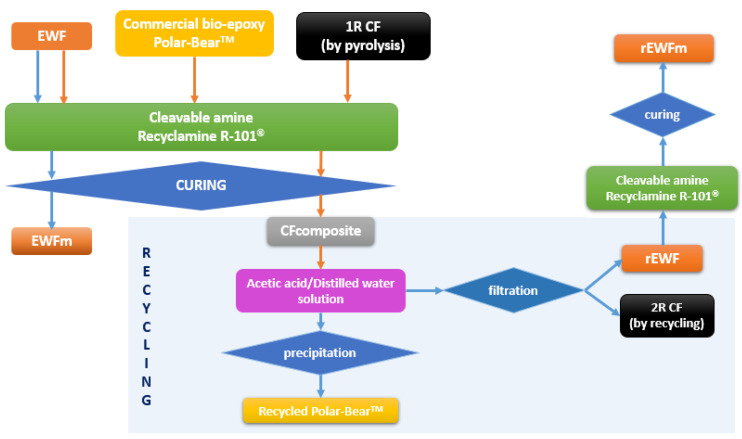
Block-flow diagram of the recycling process.

**Figure 2 polymers-14-03878-f002:**
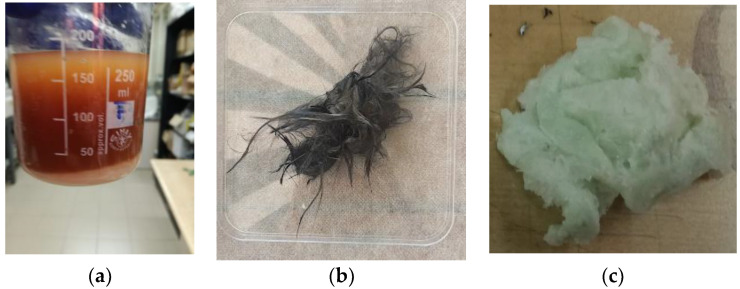
Products obtained from composite recycling: (**a**) epoxidized waste flour (EWF), (**b**) carbon fibers, and (**c**) Polar Bear^TM^ epoxy.

**Figure 3 polymers-14-03878-f003:**
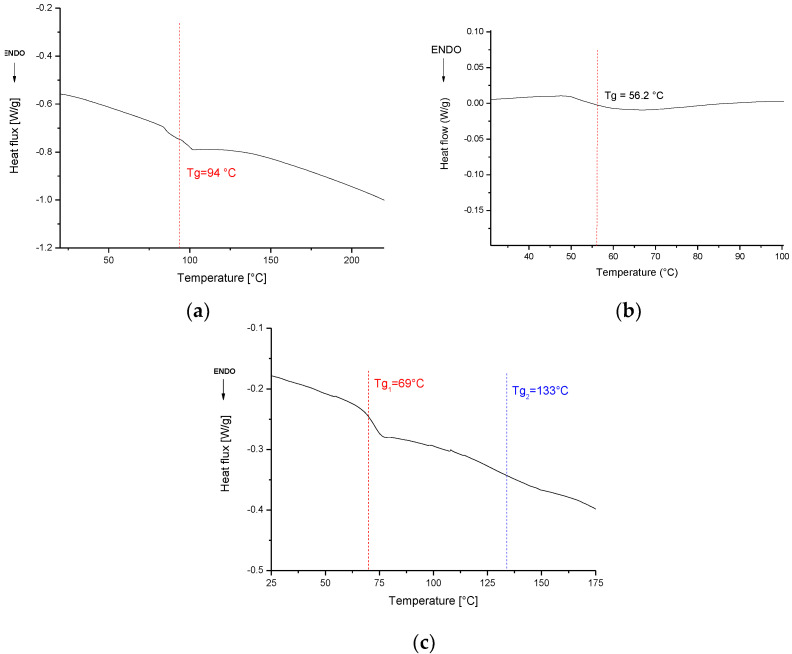
DSC: on epoxidized waste flour sample (EWFm) (**a**); on recycled EWF matrix (rEWFm) (**b**); on CF composite sample (**c**).

**Figure 4 polymers-14-03878-f004:**
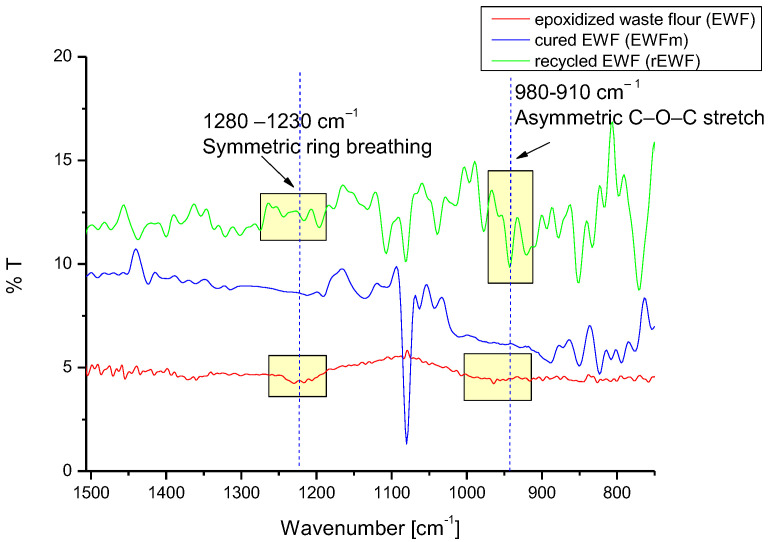
FTIR on EWF-based epoxy systems.

**Figure 5 polymers-14-03878-f005:**
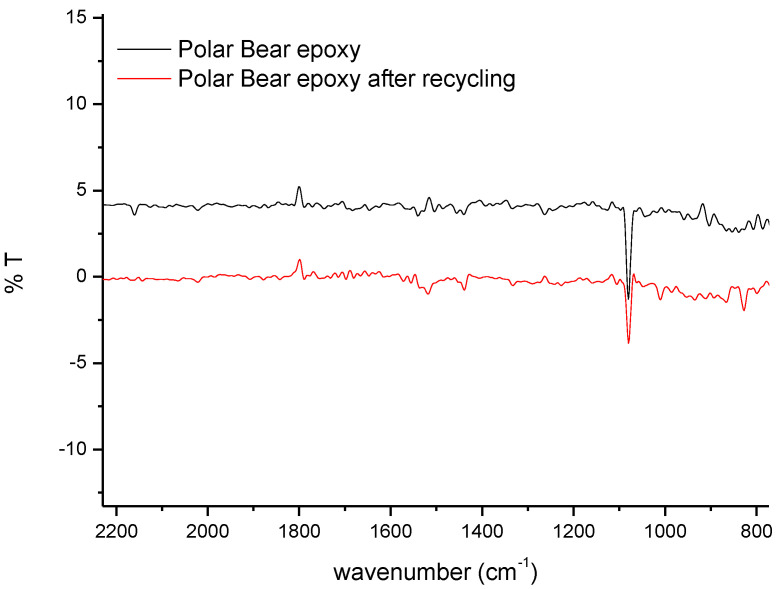
FTIR on Polar Bear^TM^ systems.

**Figure 6 polymers-14-03878-f006:**
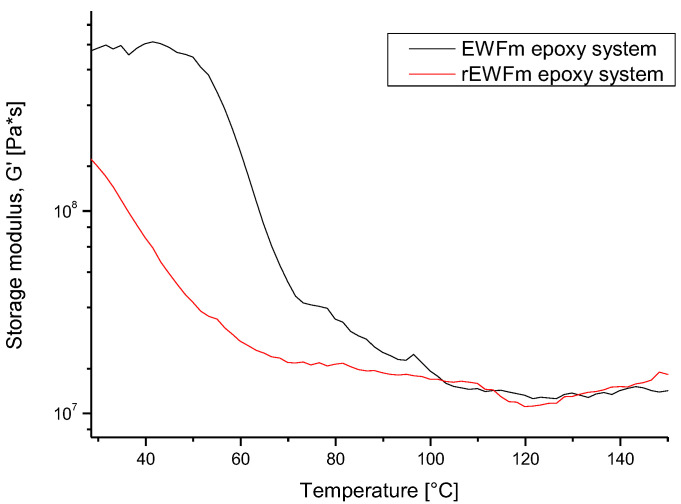
DMA on EWF-based epoxy systems.

**Figure 7 polymers-14-03878-f007:**
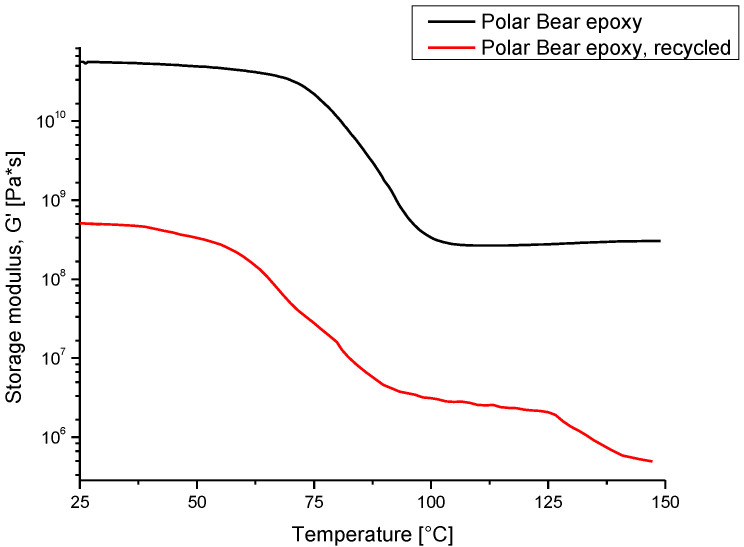
DMA on Polar Bear^TM^ epoxy systems.

**Figure 8 polymers-14-03878-f008:**
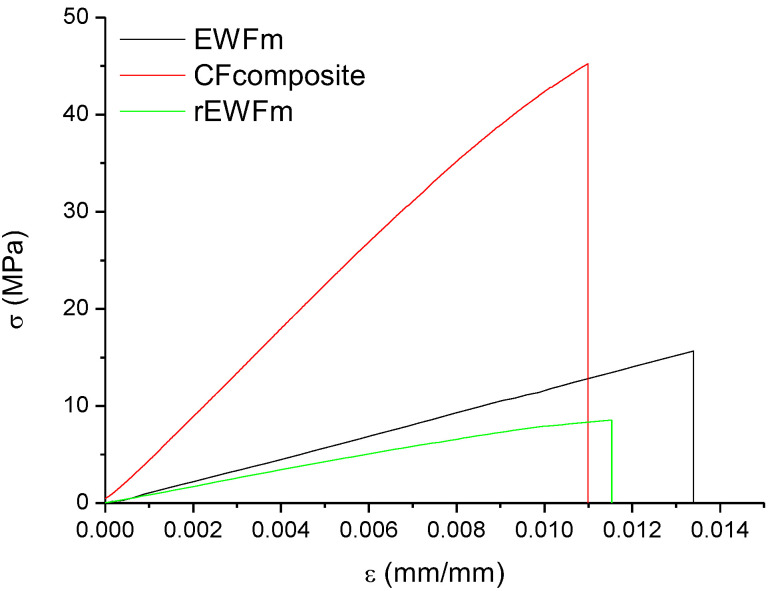
Flexural tests on EWF-based epoxy systems and composite.

**Figure 9 polymers-14-03878-f009:**
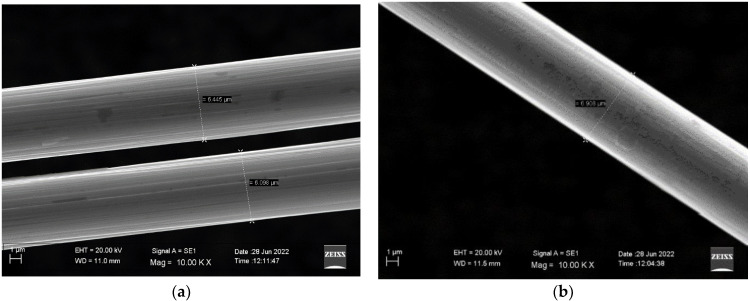
SEM analysis on carbon fibers: (**a**) *1R* and (**b**) *2R*.

**Figure 10 polymers-14-03878-f010:**
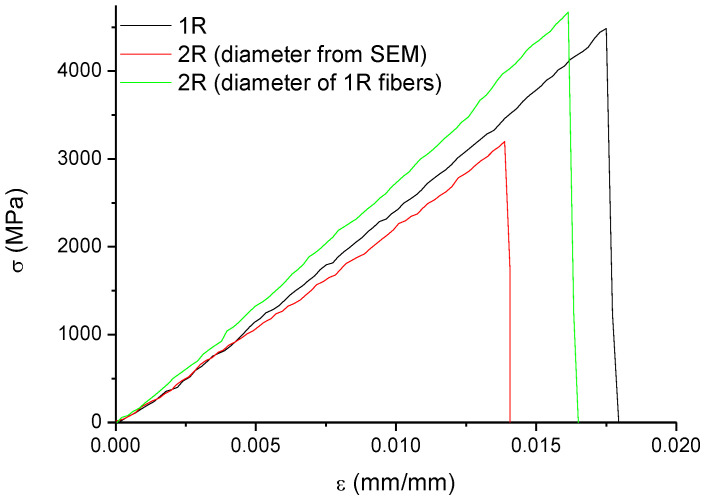
Tensile tests on *1R* and *2R* carbon fibers.

**Table 1 polymers-14-03878-t001:** Composition of EWF-matrix and composite-produced samples (as percentages).

Sample Name	Epoxidized Waste Flour (EWF) (%)	Recyclamine^TM^ R-101 (%)	Polar Bear^®^ Epoxy—Part A (%)	Recycled Carbon Fibers (*1R*) (%)
EWFm	83.33	16.66	-	-
CFcomposite	38.46	15.38	38.46	7.69
rEWFm	83.33	16.66	-	-

**Table 2 polymers-14-03878-t002:** Flexural properties of EWF-based epoxy systems and composite.

Sample Name	Flexural Strength, σ (MPa)	Strain at Break, ε (mm/mm)	Flexural Modulus, E (GPa)
EWFm	16.9 ± 5.37	0.012 ± 0.003	1.45 ± 0.54
CFcomposite	41.6 ± 17.0	0.011 ± 0.0025	3.77 ± 1.29
rEWFm	9.78 ± 1.14	0.013 ± 0.003	0.86 ± 0.068

**Table 3 polymers-14-03878-t003:** Carbon fiber diameters by SEM.

Carbon Fibers	Diameter (μm)
*2R*, recovered by solvolysis	6.908
*1R*, recycled only by pyrolysis	6.272

**Table 4 polymers-14-03878-t004:** Tensile properties of recycled carbon fibers.

Fiber Sample Name	Tensile Strength, σ (MPa)	Strain at Break, ε (mm/mm)	Tensile Modulus, E (GPa)
*1R*	3938.12 ± 777.21	0.019 ± 0.0007	269.76 ± 16.34
*2R* (diameter from SEM)	3152.25 ± 1065.16	0.015 ± 0.0010	221.88 ± 20.73
*2R* (diameter from SEM for *1R* fibers)	3836.29 ± 1297.85	0.013 ± 0.0010	271.15 ± 29.28

## Data Availability

Not applicable.

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
