# Peer review of "Innovative Closed-Loop Recyclable Bio-Based Composites from Epoxidized Waste Flour and Recycled Carbon Fibers"

_polymers, 2022, doi:10.3390/polym14183878_

Round 1

Reviewer 1 Report

The subject of article entitled “Innovative closed-loop recyclable bio-based composites from epoxidated waste flour and recycled carbon fibers” is very interesting and the article is well written and presented. However, there are one issues to be addressed before accepted for publications. namely, the authors used too little literature (11 items) to write the introduction. This would extend the introduction of this article.

Reviewer 2 Report

The manuscript entitled “Innovative closed-loop recyclable bio-based composites from 1 epoxidated waste flour and recycled carbon fibers” has been submitted by authors. Some issues to be addressed which will improve the quality of manuscript. Therefore, I recommend this work could be published after the major revision

1.      What is the novelty of this paper?

2.      The English composition requires many improvements. The authors should proofread the manuscript carefully to minimize grammatical errors.

3.      Check the format of the reference and correct all the errors.

4.      All the references mentioned in the paper should be cited in the text or vice-versa.

5.      The table and figures heading should be incorporated and discussed in the text.

6.      Fig. 6 and 7 index FTIR peak value.

7.      This research topic is widely studied in past and lot of studies are performed. Author please added comparative table for reader clear understanding.

Round 2

Reviewer 2 Report

The author solves all comments, and now it's ready to accept in its present form.